# The histone methyltransferase Setd2 is indispensable for V(D)J recombination

Zhongzhong Ji [1,2], Yaru Sheng[1], Juju Miao[1,2], Xiaoxia Li[1], Huifang Zhao[1], Jinming Wang[1], Chaping Cheng[1], Xue Wang[1,2], Kaiyuan Liu[1], Kai Zhang[1], Longmei Xu[1], Jufang Yao[1], Lijing Shen[3], Jian Hou[3], Wenhao Zhou[4], Jinqiao Sun[4], Li Li [1,2], Wei-Qiang Gao[1,2] & Helen He Zhu [1]

The diverse repertoire of T cell receptors (TCR) and immunoglobulins is generated through the somatic rearrangement of respective V, D and J gene segments, termed V(D)J recombination, during early T or B cell development. However, epigenetic regulation of V(D)J recombination is still not fully understood. Here we show that the deficiency of Setd2, a histone methyltransferase that catalyzes lysine 36 trimethylation on histone 3 (H3K36me3) in mice, causes a severe developmental block of thymocytes at the CD4−CD8− DN3 stage. While H3K36me3 is normally enriched at the TCRβ locus, Setd2 deficiency reduces TCRβ H3K36me3 and suppresses TCRβ V(D)J rearrangement by impairing RAG1 binding to TCRβ loci and the DNA double-strand break repair. Similarly, Setd2 ablation also impairs immunoglobulin V(D)J rearrangement to induce B cell development block at the pro-B stage. Lastly, SETD2 is frequently mutated in patients with primary immunodeficiency. Our study thus demonstrates that Setd2 is required for optimal V(D)J recombination and normal lymphocyte development.

[1] State Key Laboratory of Oncogenes and Related Genes, Renji-Med-X Clinical Stem Cell Research Center, and Department of Urology, Ren Ji Hospital, School of Medicine, Shanghai Jiao Tong University, Shanghai 200127, China. [2] School of Biomedical Engineering & Med-X Research Institute, Shanghai Jiao Tong University, Shanghai 200030, China. [3] Department of Hematology, Ren Ji Hospital, School of Medicine, Shanghai Jiao Tong University, Shanghai 200127, China. [4] Genetic Center, Children's Hospital of Fudan University, 399 Wanyuan Road, Shanghai 201102, China. Correspondence and requests for materials should be addressed to J.S. (email: jinqiaosun@fudan.edu.cn) or to L.L. (email: lil@sjtu.edu.cn) or to W.-Q.G. (email: gao.weiqiang@sjtu.edu.cn) or to H.H.Z. (email: zhuhecrane@shsmu.edu.cn)

In mammals, the adaptive immune response relies on diverse T cell receptors (TCRs) and immunoglobulins to recognize varied antigens[1]. The repertoire of TCRs and immunoglobulins is generated through somatic DNA rearrangements of V, D, and J gene segments during early T and B cell lymphopoiesis. T and B cell development originally initiates from hematopoietic stem cells (HSCs) that later give rise to committed lymphoid progenitors in the bone marrow (BM)[2]. Early T cell progenitors then migrate to the thymus and progress through multiple CD4$^-$CD8$^-$ double negative (DN) stages (DN1 to DN4). At the DN3 stage, TCRβ locus recombination occurs to generate functional TCRβ chains that form the pre-TCR complex with pTα and CD3 to perform signaling transduction functions and to induce further T cell differentiation into CD4$^+$CD8$^+$ double positive (DP) cells[3]. V(D)J recombination is an essential step in T cell differentiation because progenitors that are deficient in a functional pre-TCR complex will undergo programmed cell death[4]. B cell development resembles T cell development, in which V(D)J recombination of the immunoglobulin heavy chain mainly occurs at the pro-B cell stages[5].

V(D)J recombination arises in a restricted lineage- and stage-specific manner ensured by temporal production of the RAG1/2 recombinase complex and intricate epigenetic modifications of VDJ loci in early T and B cell progenitors[6–9]. The RAG1/2 complex binds to recombination signal sequences (RSSs) adjacent to V, D, and J coding regions and causes DNA double-strand breaks (DSBs). The broken coding ends are then repaired via non-homologous end joining (NHEJ) to join and generate functional TCR or immunoglobulin genes[10,11]. Therefore, the DNA damage response and repair pathways are actively involved in V(D)J rearrangement[12–14]. Epigenetic modifications are closely associated with V(D)J rearrangement, and the transcriptionally activating modification of H3K4me3 is found to directly interact with the RAG2 PHD finger domain to facilitate cleavage activity at RSSs[15–17]. Histone H3 methylation at lysine 9 and 27 and DNA methylation at CpG sites, two transcriptionally suppressive modifications, are preferentially enriched in the recombinationally inactive state[18,19]. However, our understanding of the epigenetic mechanisms that regulate V(D)J recombination is still highly incomplete. In addition, the role of enzymes that catalyze epigenetic modifications in V(D)J recombination needs further elucidation.

Trimethylation of histone H3 at lysine 36 has been recently demonstrated to be a pivotal epigenetic modification in DNA repair: it promotes DNA mismatch repair by directly interacting with Mut5α[20] and facilitates double-streak break repair by recruiting 53BP1 to DSB sites[21]. H3K36me3 is also frequently associated with active transcription by participating in pre-mRNA elongation and splicing[22]. SETD2 is the sole trimethyltransferase responsible for H3K36me3 and is frequently mutated in various types of malignancies in lymphoid cell lineages, including acute lymphoblastic leukemia[23,24], enteropathy-associated T cell lymphoma (EATL), and hepatosplenic T cell lymphoma (HSTL)[25,26]. However, whether Setd2 is dispensable for lymphopoiesis is not known.

Here, we generate Setd2 conditional knockout mice to study the functions of Setd2 and Setd2-mediated H3K36me3 modification during lymphocyte development. We find that Setd2 deficiency causes severe blocks in T and B cell lymphopoiesis due to the impaired V(D)J rearrangement. Moreover, we further identify that loss of Setd2 lead to the decreased accessibility of RAG1 to antigen receptor gene loci and insensitivity of DSB repair response in early lymphocytes.

## Results

**High Setd2 expression in mouse lymphoid lineage.** We first assessed the expression levels of Setd2 in different compartments of the hematopoietic system. HSCs, progenitors, and differentiated hematopoietic cells from bone marrow or thymus were sorted by flow cytometry for RNA extraction and qRT-PCR. As shown in Fig. 1a, Setd2 was preferentially expressed in HSCs, common lymphoid progenitors (CLPs), megakaryocyte/erythroid progenitors (MEPs), and early T and B progenitors, with the highest expression detected in CLPs; in contrast, the myeloid lineage exhibited a relatively lower mRNA level of Setd2. The preferentially high expression of Setd2 in hematopoietic and lymphoid stem/progenitor cells implies that Setd2 may affect hematopoiesis and lymphopoiesis.

**Generation of the Setd2 conditional deletion mouse model.** To decipher the function of Setd2 in hematopoiesis, we generated a *Setd2$^{f/f}$* genetically modified mouse line (Fig. 1b), in which exon 6 and exon 7 of Setd2 were flanked by the loxP element. *Setd2$^{f/f}$* mice were crossed with *Mx1-Cre* transgenic mice to obtain conditional hematopoietic knockout mice. Two weeks after the final injection of 3 doses of poly(I:C) (pIpC), we detected efficient deletion of Setd2 expression in bone marrow nucleated cells (BMNCs) from *Mx1-Cre$^+$;Setd2$^{f/f}$* mice (Fig. 1c). Consistent with the observation that Setd2 is the major histone methyltransferase that catalyzes the trimethylation of lysine 36 on histone 3[23], H3K36me3 was barely detectable in Setd2 knockout BMNCs (Fig. 1c, d), while H3K36me2 was not affected by loss of Setd2 (Fig. 1c).

**Setd2-deficient mice have less mature B and T cells.** We next performed a complete blood cell count (CBC) on the peripheral blood (PB) of *Mx1-Cre$^+$;Setd2$^{f/f}$* and control mice at 8 weeks post pIpC treatment. As shown in Supplementary Table 1, the monocyte and red blood cell counts from *Mx1-Cre$^+$;Setd2$^{f/f}$* mice exhibited a slight decrease, while the platelet counts exhibited a moderate increase. The most prominent effect of Setd2 loss on the CBC was observed for white blood cells (WBCs) and lymphocytes. We observed a marked reduction of WBC and lymphocyte counts in Setd2 knockout mice compared to these counts in controls (Fig. 1e). Flow cytometric analysis further demonstrated significant decreases in the CD3e$^+$ T cell and B220$^+$ B cell counts in the peripheral blood of *Mx1-Cre$^+$;Setd2$^{f/f}$* mice (Fig. 1f, g). Consistent with these results, the counts of BMNCs and bone marrow lymphocytes were significantly decreased in *Mx1-Cre$^+$; Setd2$^{f/f}$* mice (Fig. 1h–j) Taken together, these findings suggest that Setd2 is actively involved in lymphoid lineage differentiation.

**Deficient HSC capacity but increased CLP in Setd2 knockout.** Mature lymphocytes in mammals are differentiated through multiple progenitor stages from rare HSCs. To explore the cause of the lymphopenia phenotype in *Mx1-Cre$^+$;Setd2$^{f/f}$* mice and to determine which step of lymphocyte differentiation was affected by *Setd2* knockout, we further performed FACS analysis of HSCs and committed progenitors. We found a decrease in the HSC-enriched Lin$^-$Sca1$^+$Kit$^+$ (LSK) cell population (Fig. 2a–c). However, the CLP population exhibited an evident increase after ablation of Setd2 (Fig. 2d–f). To further examine the impact of Setd2 ablation on hematopoiesis under stress, we performed bone marrow transplantation experiments. BMNCs were harvested from untreated *Mx1-Cre$^+$;Setd2$^{f/f}$* or littermate control mice and mixed at a 1:1 ratio with BMNCs from CD45.1 mice before bone marrow transplantation into lethally irradiated animals. Four weeks after transplantation, recipients received three doses of pIpC injection to induce Setd2 knockout. Beginning 2 weeks after the last injection, we examined the peripheral blood of recipient mice monthly to evaluate the contribution of Setd2-deficient or control bone marrow. As

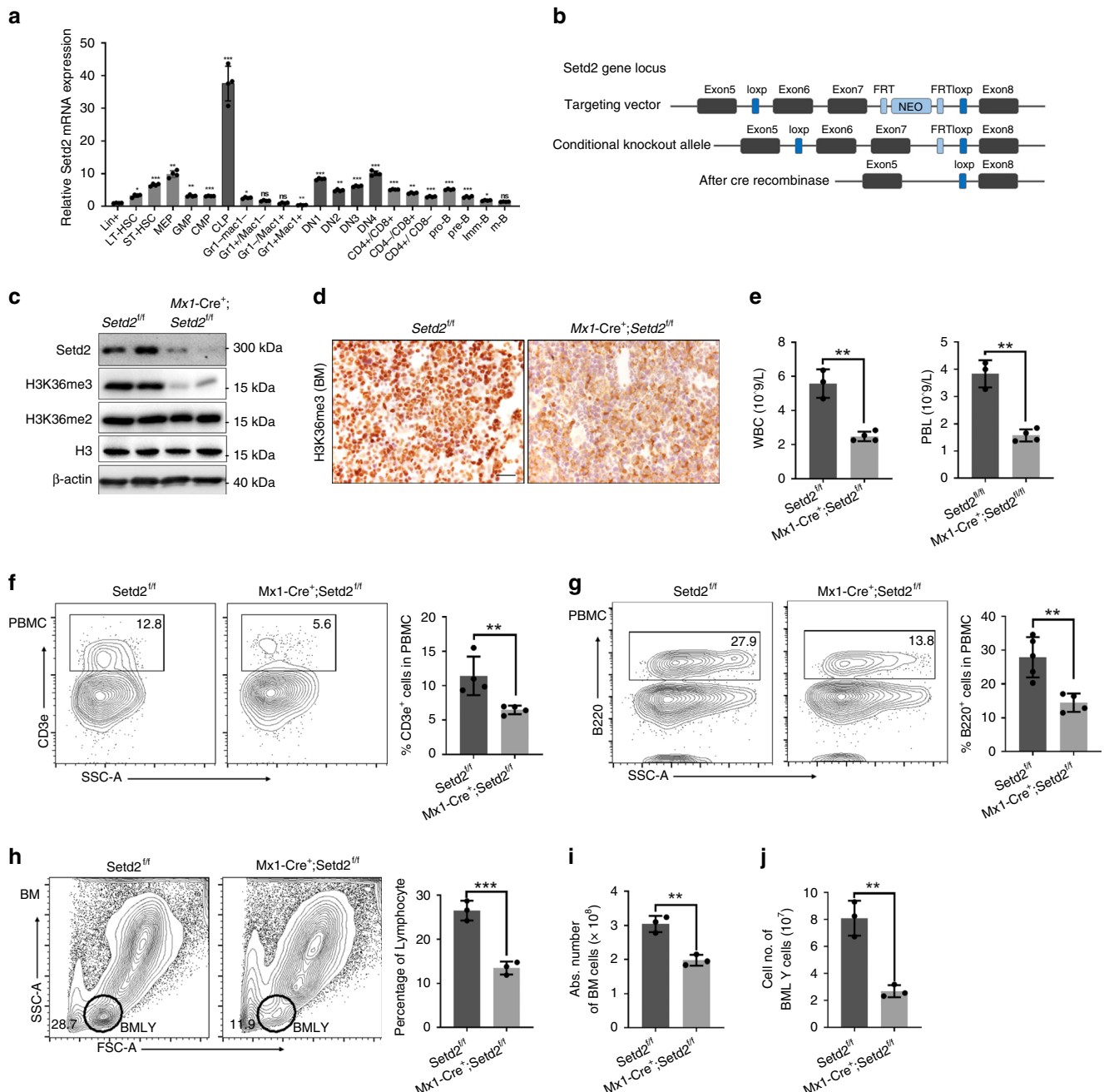

**Fig. 1** Setd2-deficient mice exhibit reduced numbers of T and B lymphocytes. **a** Relative mRNA expression of Setd2 in different hematopoietic cell compartments ($n = 4$). **b** Schematic of the strategy for establishing the *Setd2^{f/f}* mouse line. **c** Immunoblotting of Setd2 and H3K36me3 in bone marrow nucleated cells (BMNCs) from Setd2 knockout mice. H3 and β-actin were used as the loading controls. **d** Immunohistochemistry of H3K36me3 in femur sections from Setd2 knockout and control mice. **e** Complete blood count of peripheral blood showed lymphopenia in pIpC-treated *Mx1-Cre^+;Setd2^{f/f}* mice. (WBC, white blood cell; PBL, peripheral blood lymphocyte; $n = 7$). **f, g** Flow cytometric analysis of CD3e⁺ T cells (**f**) and B220⁺ B cells (**g**) in the peripheral blood of Setd2 knockout and control mice ($n = 8$–9). **h–j** Flow cytometric analysis of bone marrow and bone marrow lymphocytes (BMLYs) in Setd2 knockout and control mice. ($n = 6$). (Data were collected at 8 weeks after the final pIpC injection. The data are presented as the means ± SDs. *$p < 0.05$; **$p < 0.01$; ***$p < 0.001$; ns, not significant, Source data are provided as a Source Data file)

shown in Fig. 2g, compared to the expected 50% of peripheral blood cells generated by control BMNCs, a significantly lower percentage of peripheral blood cells was derived from Setd2 knockout BMNCs. In addition, analysis of the bone marrow of recipient mice showed a consistent reduction in the generation of BMNCs from Setd2-deficient cells (Fig. 2h, i), suggesting that Setd2 deletion caused a severe blood cell repopulation disadvantage. Moreover, we observed that significantly fewer LSK

cells in recipient animals were derived from Setd2 knockout BM transplants (Fig. 2j). Phenotypically, similar to the Setd2 knockout donor mice, the Setd2-deficient recipients exhibited robust expansion of CLPs (Fig. 2k) but impaired reconstitution of T cells and B cells in the BM and peripheral blood (Supplementary Fig. 1). Collectively, these results indicate that the ablation of Setd2 functionally impairs HSCs and disturbs lymphoid lineage differentiation.

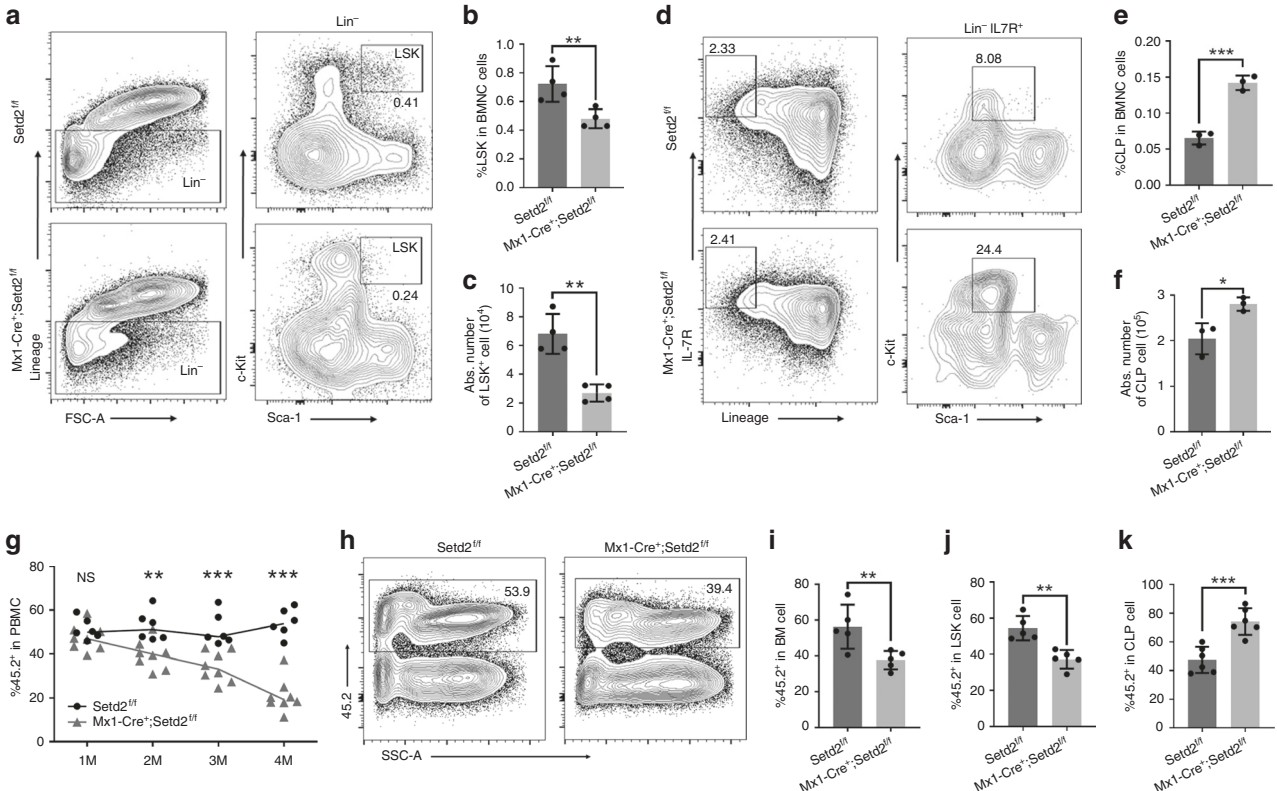

**Fig. 2** Setd2 knockout leads to decreased HSC numbers and repopulating capacity and to skewed lymphoid differentiation. **a–c** The flow cytometry profiles of LSK (Lineage−Sca-1+C-kit+) cells in bone marrow from Setd2 knockout and control mice show a decrease in the frequency (**b**) and total number (**c**) of HSC-enriched LSK cells (n = 8). **d–f** Representative flow cytometry plots of common lymphoid progenitors in bone marrow from Setd2 knockout and control mice (**d**); the frequency and absolute number of CLPs are shown in (**e**) and (**f**). (CLP: Lineage−IL-7R+c-kit+Sca-1mid; n = 6). **g** Peripheral blood chimerism was examined monthly after pIpC treatment of recipient mice (n = 14). **h–k** Flow cytometric analysis of the frequency of CD45.2-derived BMNCs (**h**, **i**), LSK cells (**j**), and CLPs (**k**) in recipients at 4 months after pIpC induction (n = 10). (The data are presented as the means ± SDs. *p < 0.05; **p < 0.01; ***p < 0.001; ns, not significant, Source data are provided as a Source Data file)

**Ablation of Setd2 blocks T cell development at the DN3 stage.** The accumulation of CLPs and reduction in mature peripheral blood T cells in Setd2-deficient mice suggested a blockage of T cell differentiation. We therefore further investigated at which step the T cell lineage differentiation was impaired by Setd2 knockout. Early T cell progenitors migrate from the BM to the thymus and progress through multiple DN stages, namely, DN1 (CD44+CD25−CD4−CD8−), DN2 (CD44+CD25+CD4−CD8−), DN3 (CD44−CD25+CD4−CD8−), and DN4 (CD44−CD25−CD4−CD8−)[3]. As shown in Fig. 3a, Setd2-deficient mice exhibited a visibly smaller thymus, in which the total thymic cellular numbers were approximately 4-fold lower than those in control mice (Fig. 3b). The thymus from Setd2 knockout mice histologically displayed an altered architecture that lacked the well-defined medullary and cortical structures seen in control mice (Fig. 3a). We next analyzed thymocytes by FACS to assess the impact of Setd2 loss on the DN, DP, and SP populations. We found that in the thymus of pIpC-treated *Mx1-Cre+;Setd2f/f* mice, the percentage of DN cells was markedly elevated, while the percentage of DP cells was significantly reduced (Fig. 3c–e). Further examination of DN cells revealed that in the Setd2-deficient mice, the subpopulations of DN1 and DN3 thymocytes were dramatically enlarged, whereas the subpopulation of DN4 thymocytes was significantly decreased (Fig. 3f–h), suggesting that thymocyte differentiation was developmentally blocked at the transition from the DN3 to the DN4 stage.

To exclude a possible effect of the microenvironment of the bone marrow and other tissues on the differentiation of lymphoid

progenitors in the *Mx1-Cre+;Setd2f/f* mouse model, we crossed *Setd2f/f* mice with *Lck-Cre* mice to achieve specific deletion of Setd2 in the T cell lineage. Phenotypically consistent with *Mx1-Cre+;Setd2f/f* mice, *Lck-Cre+;Setd2f/f* mice exhibited significantly reduced thymus sizes, decreased thymocyte numbers (Fig. 4a, b) and altered histology (Fig. 4c). Similarly, compared with control mice, *Lck-Cre+;Setd2f/f* mice displayed reductions in bone marrow and peripheral blood T cells (Fig. 4d, e), increased DN thymocytes but decreased DP thymocytes (Fig. 4f–h), and an apparent developmental block at the DN3 stage (Fig. 4i–k), suggesting that Setd2 plays a pivotal role in lymphocyte development. Collectively, these data suggest that Setd2 ablation results in T cell lymphopenia due to blockage of lymphocyte differentiation at the DN3 stage.

**Setd2 is required for TCRβ rearrangement in thymocytes.** During T lymphocyte development, V(D)J rearrangement, which is mediated by RAG recombinases at the DN stage for the generation of a diverse TCRβ repertoire, is a critical checkpoint event during progression from DN to DP[11,27,28]. Deficiency of RAG recombinases leads to arrested lymphocyte development at the pro-T and pro-B stages and an absence of mature T or B cells in RAG−/− mice[6,7]. Previous reports revealed that histone hypermethylation is associated with V, D, and J loci rearrangement activity[15,29]. Considering the phenotype of the Setd2-deficient T lymphocyte developmental block at the DN stage when V(D)J rearrangement occurs and the observation that Setd2 is the major

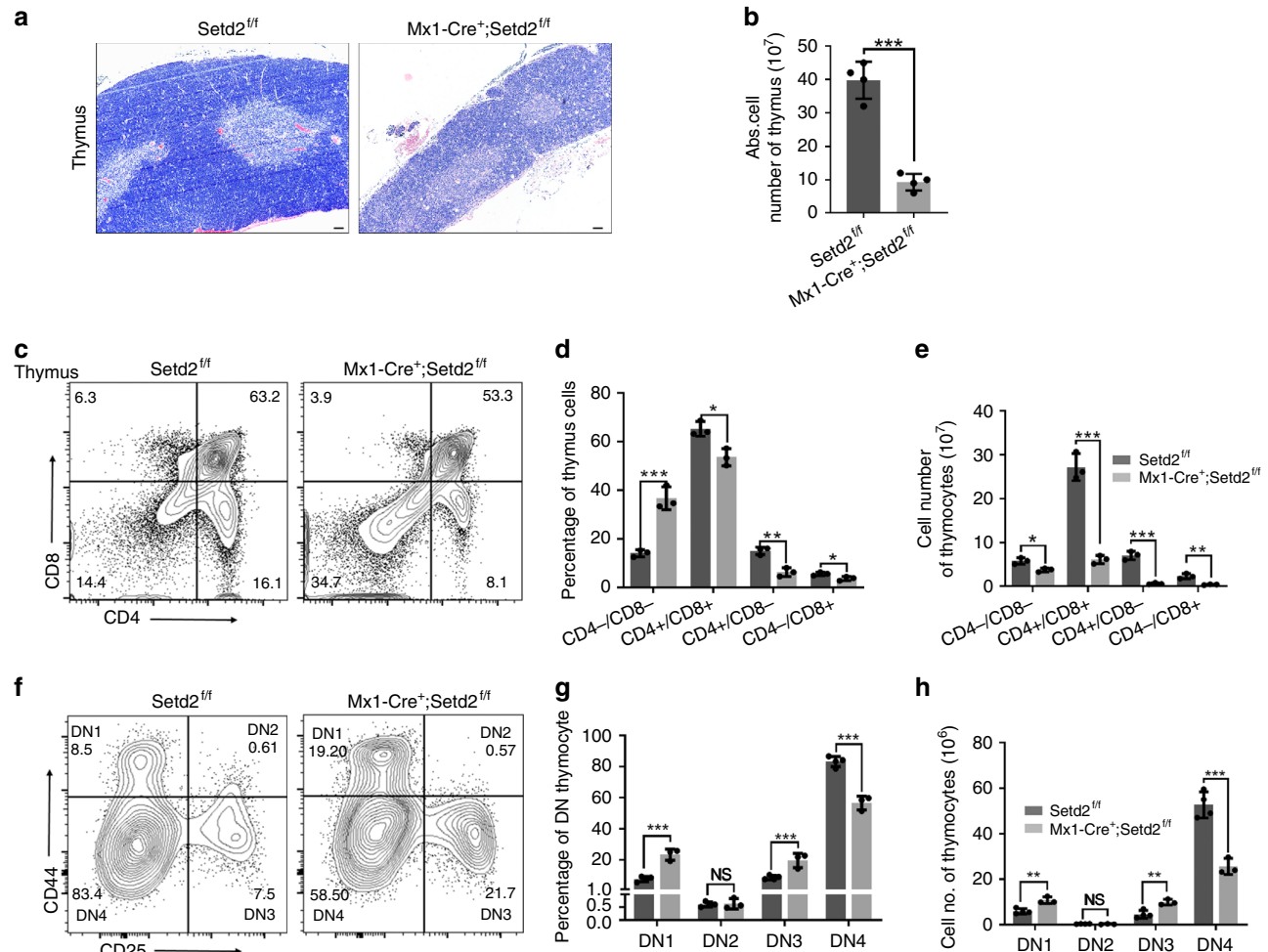

**Fig. 3** T cell development is blocked at the DN3 stage in *Mx1-Cre+;Setd2f/f* mice. **a** Representative hematoxylin–eosin (H&E) staining of thymus sections from Setd2-deficient and control mice (scale bar = 50 μm). **b** Total thymic cell number in Setd2-deficient mice and control mice ($n = 8$). **c**–**e** Representative flow cytometry plots of anti-CD4 and anti-CD8 antibody staining in thymocytes (**c**) and the frequency (**d**) and cell numbers (**e**) of subpopulations of thymocytes from Setd2-null and control mice after pIpC treatment ($n = 6$). **f**–**h** Flow cytometric analysis showing subpopulations of double-negative, DN1 (CD44+CD25−), DN2 (CD44+CD25+), DN3 (CD44−CD25+), and DN4 (CD44−CD25−) thymocytes from Setd2-deficient and control mice ($n = 6$). (The data are presented as the means ± SDs. *$p < 0.05$; **$p < 0.01$; ***$p < 0.001$; ns, not significant, Source data are provided as a Source Data file)

H3K36me3 histone methyltransferase, we hypothesized that H3K36me3 might be required for V(D)J rearrangement. To test this hypothesis, we first performed a ChIP assay to examine the distribution of H3K36me3 at the TCRβ locus using sorted DN3 thymocytes from 10-week-old C57BL/6 mice. Paired primers that were close to or from the RSSs were chosen for individual V, D, and J segments. Paired primers for the intergenic regions between the V cluster and D cluster were also included. ChIP analysis revealed significant enrichment of H3K36me3 in the Vβ, Dβ, and Jβ segments located at or close to the RSSs of the TCRβ gene locus in DN3 thymocytes, preferentially in proximal and distal Vβ segments and Dβ regions, relative to that in the input. (Fig. 5a). Additional ChIP assays showed that Setd2 deletion led to removal of H3K36me3 deposition at TCRβ locus in thymocytes from *Lck-Cre+; Setd2f/f* mice (Supplementary Fig. 2a). We next examined the rearrangement at the TCRβ locus of DN3 thymocytes using genomic DNA from the sorted DN3 thymocytes from *Lck-Cre+; Setd2f/f* or control mice. We found that Dβ-to-Jβ rearrangement, including Dβ1-to-Jβ1 and Dβ2-to-Jβ2, was impaired in Setd2-deficient mice (Fig. 5b). TRBV30-to-DJβ (including TRBV30-to-DJβ1.7 and TRBV30-to-DJβ2.7) and TRBV12-2-to-DJβ (including TRBV12-2-to-DJβ1.7 and TRBV12-2-to-DJβ2.7) rearrangements,

which represent proximal and near-distal V(D)J recombination, respectively, were dramatically decreased in DN3 thymocytes from *Lck-Cre+;Setd2f/f* mice compared to these rearrangements in control mice. In addition, TRBV3-to-DJβ (including TRBV3-to-DJβ1.7 and TRBV3-to-DJβ2.7) recombination was barely detectable in *Lck-Cre+;Setd2f/f* DN3 thymocytes (Fig. 5c, d). Interestingly, Setd2 deletion also led to an impairment of V(D)J recombination at the TCRα locus, although the effect on TCRα recombination was not as profound as that on TCRβ gene recombination (Supplementary Fig. 3).

To further test whether Setd2 is indispensable for V(D)J gene rearrangement of TCRβ in T cell progenitors, we generated a retrogenic TCR mouse model by introducing the TCR (coexpressed with GFP) to BM cells via retroviral transfection for a rescue experiment (Supplementary Fig. 4a)[30]. We first examined the deletion of Setd2 using purified genomic DNA from GFP+ thymocytes of the recipients and found that the GFP+ (representing the transgenic TCR) thymocytes were derived from the *Lck-Cre+;Setd2f/f* donor mice with efficient ablation of Setd2 in thymocytes (Supplementary Fig. 4b). We found that the expression of TCR in *Setd2−/−* T cells significantly abrogated the developmental block of DN3 to DN4 progression caused by Setd2

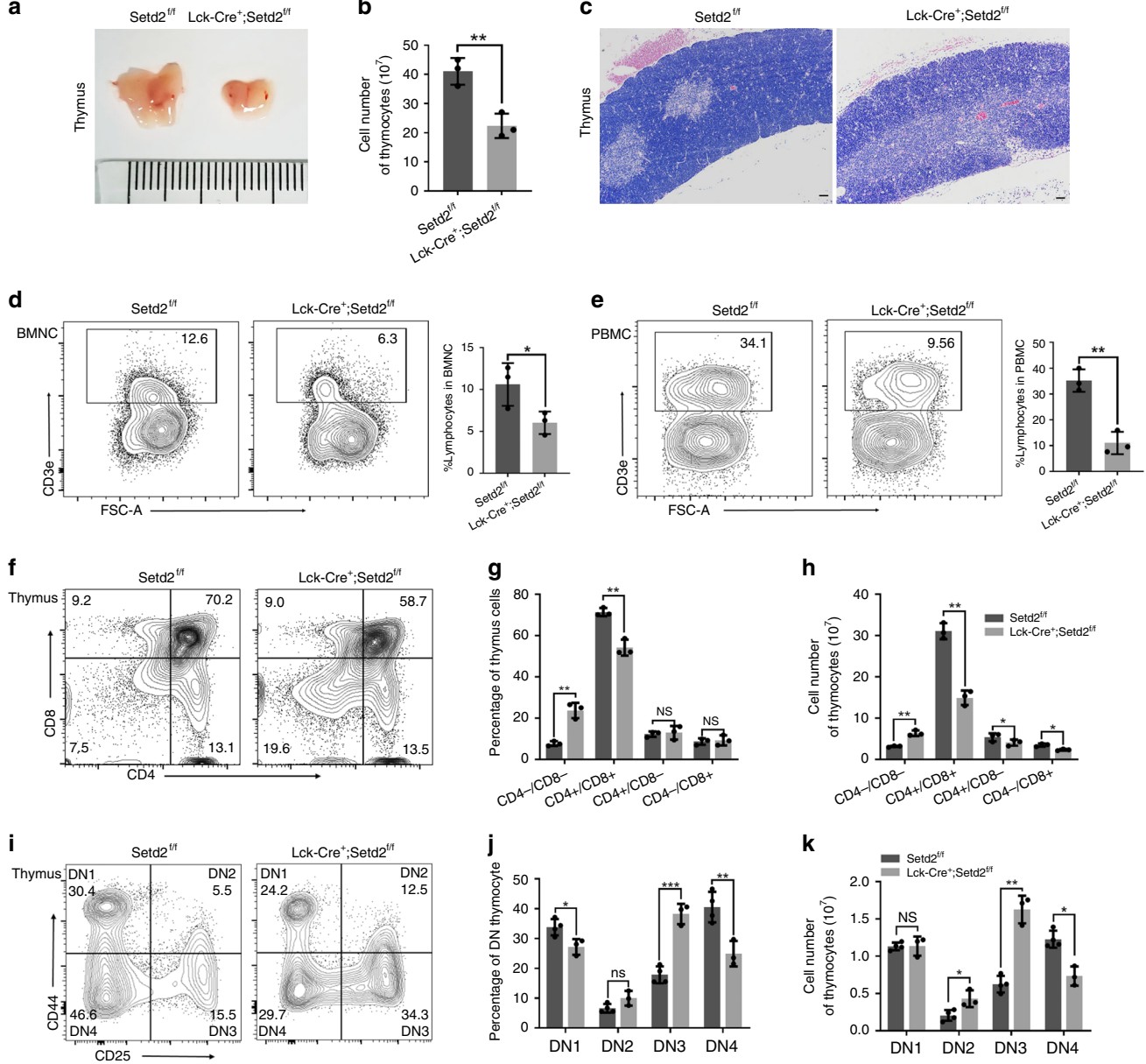

**Fig. 4** Lymphoid-specific deletion of Setd2 leads to a block of T cell development at the DN3 stage. **a** *Lck-Cre+;Setd2f/f* mice exhibit a visibly smaller thymus than control mice (unit of ruler = 1 mm). **b** The thymocyte cell number was significantly lower in *Lck-Cre+;Setd2f/f* mice than in control mice (*n* = 6). **c** Representative hematoxylin–eosin (H&E)-stained thymus sections from *Lck-Cre;Setd2fl/fl* and control mice (scale bar = 50 μm). **d**, **e** Flow cytometric analysis of CD3e+ T cell populations in bone marrow mononuclear cells (BMNCs) (**d**) and peripheral blood mononuclear cells (PBMCs) (**e**) indicated a reduced CD3e+ T cell population in *Lck-Cre+;Setd2f/f* mice (*n* = 6). **f–h** Flow cytometric analysis of subpopulations of double negative T cells in *Lck-Cre+; Setd2f/f* mice and control mice with CD4 and CD8 staining (*n* = 6). **i–k** Flow cytometric analysis indicated that the subpopulations of DN2 (CD44−CD25+) and DN3 (CD44−CD25+) thymocytes were significantly increased, whereas the DN4 (CD44−CD25−) thymocyte population was reduced, in *Lck-Cre+; Setd2f/f* mice (*n* = 6). (The data are presented as the means ± SDs. *$p < 0.05$; **$p < 0.01$; ***$p < 0.001$; ns, not significant, Source data are provided as a Source Data file)

deletion (Supplementary Fig. 4c). These data provide further support for the conclusion that Setd2 is required for V(D)J recombination in normal lymphocyte development.

**Setd2 deletion also inhibits Igh rearrangement in B lineage.** The process of V(D)J recombination of the immunoglobulin heavy chain (Igh) resembles that of TCRβ gene rearrangement[5]. We then sought to determine whether Setd2 was also involved in Igh gene rearrangement. pro-B cells (B220+CD43+CD19+IgM−) sorted from the bone marrow of *Mx1-Cre+; Setd2f/f* and control

mice at 8 weeks after the final injection of pIpC were harvested to assess V(D)J rearrangement. As shown in Supplementary Fig. 5a, the $V_HJ558$-to-$DJ_H$, $V_H7183$-to-$DJ_H$, and $V_HQ52$-to-$DJ_H$ rearrangements, which represent distal and proximal V(D)J recombination of the Igh gene, respectively, were markedly reduced after ablation of Setd2. Consistently, we found that B cell development was blocked at the pro-B (B220+CD43+CD19+IgM−) stage, where Igh rearrangement occurs, in Setd2 knockout bone marrow (Supplementary Fig. 5b–d). The number of CD19+IgM+ splenic B cells also exhibited a significant decrease after Setd2 ablation (Supplementary Fig. 5e).

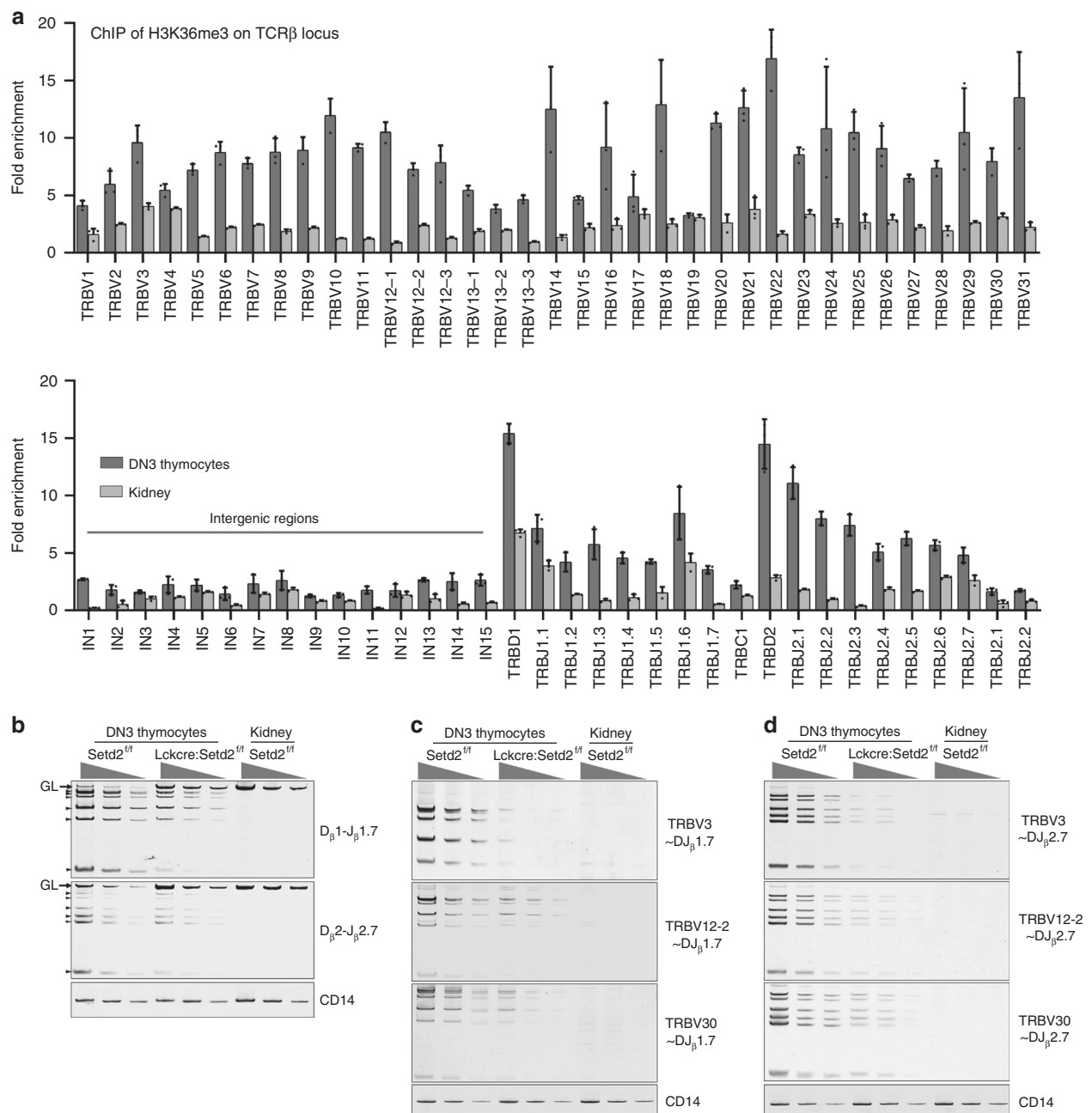

**Fig. 5** Ablation of Setd2 impairs V(D)J rearrangement at TCRβ gene loci. **a** ChIP analysis of H3K36me3 at TCRβ gene loci in DN3 thymocytes. Mouse kidney was used as the negative control. Fold enrichments are normalized to input DNA. **b** Semi-quantitative PCR analysis of the indicated $D_\beta$-to-$J_\beta$ segment rearrangements in sorted DN3 thymocytes from *Lck-Cre+;Setd2f/f* and control mice. **c, d** Semi-quantitative PCR analysis of the indicated **c** $V_\beta$-to-$DJ_\beta1.7$ and **d** $V_\beta$-to-$DJ_\beta2.7$ rearrangements in sorted DN3 thymocytes from *Lck-Cre+;Setd2f/f* and control mice. TRBV3 represents the distal Vβ segment, while TRBV12-2 and TRBV30 were selected to represent the middle and proximal Vβ segments, respectively. (For (**b–d**), mouse kidney was used as the negative control. GL: germline, black arrows: $D_\beta2/J_\beta2.1$-to-$D_\beta2/J_\beta2.6$ rearrangement. CD14 was used as the loading control. DNA was diluted to 3 concentrations ranging from 90 ng to 10 ng and subjected to 31 amplification cycles for V(D)J or D–J joint amplification or 25 cycles for CD14 amplification, Source data are provided as a Source Data file)

We further crossed *Setd2f/f* mice with *Cd19-Cre* transgenic mice to achieve specific knockout of Setd2 in the B cell lineage. H3K36me3 modification was significantly reduced in sorted B220+CD19+ B cells from *Cd19-Cre+;Setd2f/f* mice compared with that in control mice (Supplementary Fig. 6a), suggesting efficient Setd2 deletion in B cells of *Cd19-Cre+;Setd2f/f* mice. We analyzed B cell development in *Cd19-Cre+;Setd2f/f* and control mice by flow cytometry. Consistent with our observations in *Mx1-Cre+;Setd2f/f* mice, B cell development was blocked at the

pro-B stage (Supplementary Fig. 6b). V(D)J recombination of Igh in pro-B cells from *Cd19-Cre+; Setd2f/f* mice was impaired upon Setd2 deletion (Supplementary Fig. 6c). Collectively, these findings indicate that Setd2 is required for V(D)J rearrangement of Igh and for B cell development.

**Setd2 deficiency reduces Rag1-RSS binding and DNA repair.** To understand the molecular mechanism underlying the V(D)J recombination defects induced by Setd2 loss, we first examined

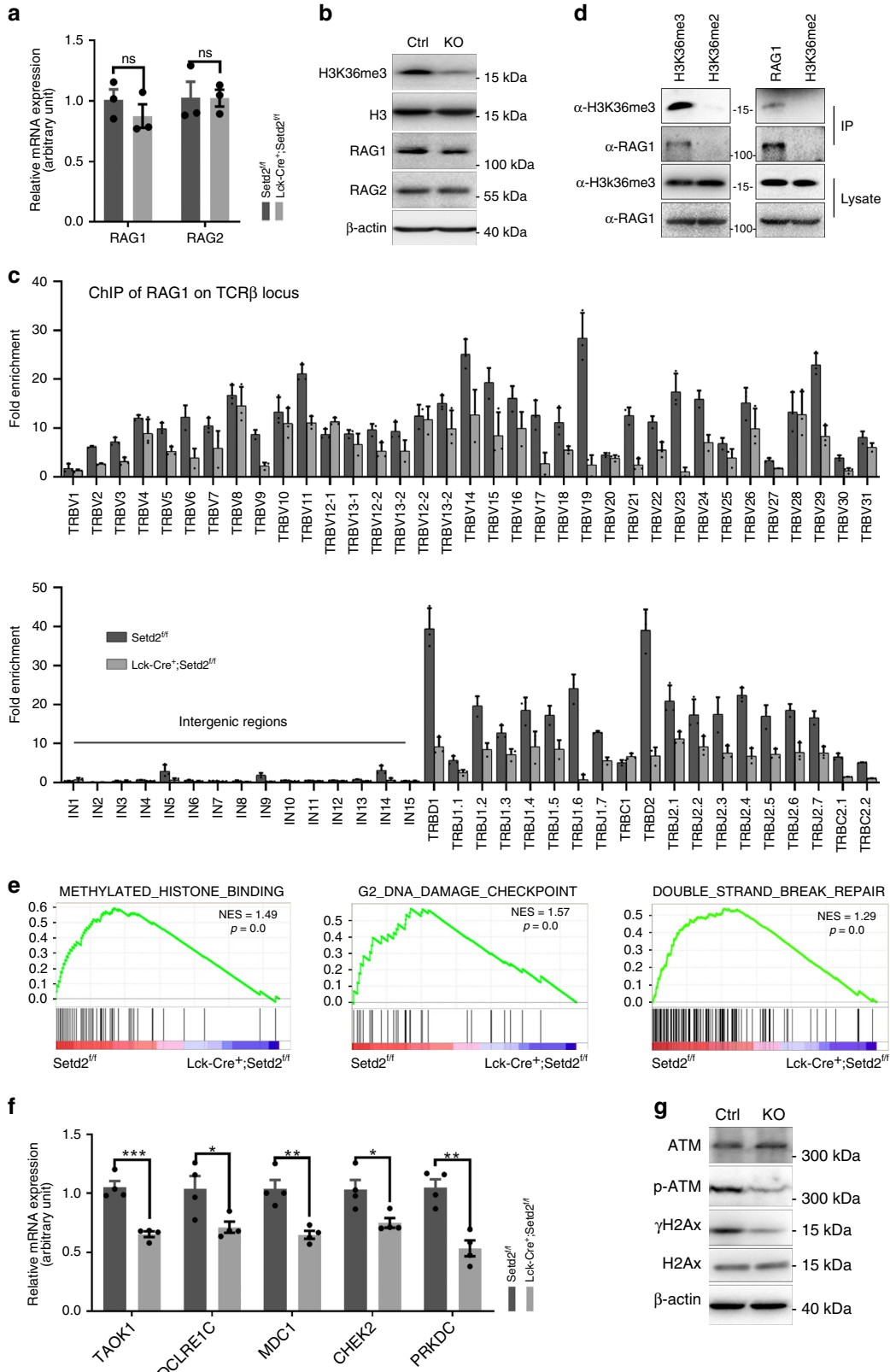

the expression levels of two essential recombinases for V(D)J rearrangement, Rag1 and Rag2. As shown in Fig. 6a, b, neither the mRNA nor protein levels of *Rag1* and *Rag2* were affected by Setd2 knockout. We then investigated whether the binding of Rag1 to the TCRβ gene was altered by the removal of H3K36me3 in thymocytes. ChIP assays revealed that Rag1 enrichment at the

TCRβ locus was reduced in the V, D, and J segments in *Lck-Cre; Setd2^{f/f}* DN thymocytes compared to that in controls (Fig. 6c). Interestingly, we noted a similar distribution of H3K36me3 and Rag1 at the TCRβ locus (Figs. 5a and 6c). We further used the Pearson correlation coefficient[31] to measure the correlation between H3K36me3 and Rag1 deposition at RSSs. We found a

**Fig. 6** Setd2 deletion in thymocytes impairs Rag1 binding to the TCRβ gene and DNA DSB repair. **a, b** The relative mRNA expression (**a**) or protein expression (**b**) levels of Rag1 and Rag2 from *Lck-Cre+;Setd2f/f* mice with Setd2 knockout. H3 and β-actin were used as the loading controls for immunoblotting (n = 6). (The data are presented as the means ± SEM. ns, not significant). **c** ChIP analysis of Rag1 at V, D, and J segments of the TCRβ gene in *Lck-Cre+; Setd2f/f* DN thymocytes. **d** Co-IP analysis of interaction between endogenous Rag1 and H3K36me3 in mouse thymocytes (the experiments were repeated at least three times). **e** GSEA analysis of RNA-Seq data for methylated histone binding, G2 DNA damage checkpoint and non-recombinational DNA repair in DN3 thymocytes from *Lck-Cre+;Setd2f/f* mice and control mice. **f** Relative mRNA expression of genes from RNA-seq data of DN3 thymocytes from *Lck-Cre+;Setd2f/f* mice and control mice. (n = 8). **g** Immunoblotting showing phosphorylation of ATM and H2Ax in Setd2-null and wild-type thymocytes (ATM, H2Ax, β-actin (as the loading control)). (The data are presented as the means ± SEM. *p < 0.05; **p < 0.01; ***p < 0.001; ns, not significant, Source data are provided as a Source Data file)

correlation *r* value of 0.7898 and a *p* value of 0.010 between H3K36me3 and Rag1 deposition, which indicated a strong positive correlation. The coimmunoprecipitation experimental results indicated that Rag1 interacted with H3K36me3 in thymocytes (Fig. 6d), implying that H3K36me3 might be involved in the recruitment of Rag1 to the RSSs during V(D)J recombination.

We also performed a ChIP assay to examine H3K36me3 and Rag1 deposition at Igh loci from sorted B220+CD19+ B cells. We found a similar overlapping pattern of H3K36me3 and RAG1 deposition at Igh loci in *Setd2f/f* mice (r = 0.7349, p < 0.001) (Supplementary Fig. 7a, b). Specific Setd2 knockout in B cells of *Cd19-Cre+; Setd2f/f* mice resulted in a reduction of RAG1 binding to Igh loci (Supplementary Fig. 7a, b). Collectively, these data suggested that the V(D)J rearrangement deficiency in Setd2-null lymphocytes was not due to downregulation of Rag1 or Rag2 expression but rather, at least partially, to decreased occupancy of Rag1 on the TCRβ or Igh genes and subsequent insufficient cleavage at RSSs mediated by the Rag1/2 complex.

Previous reports demonstrated that interaction between Rag2 and H3K4me3 is required for Rag2 recruitment to the RSSs and efficient V(D)J recombination[16,17,32,33]. We found that the H3K4me3 level was not altered by *Setd2* knockout in T cells (Supplementary Fig. 8a). Moreover, we performed ChIP assays of H3K4me3 at the TCRβ and Igh loci and did not find significant reduction in H3K4me3 deposition at the TCRβ and Igh genes by Setd2 knockout (Supplementary Fig. 8b, c). These data suggest that the V(D)J recombination defect in Setd2 knockout lymphocytes did not result from altered H3K4me3.

We further performed RNA-Sequencing using RNA extracted from FACS-sorted DN3 thymocytes from *Lck-Cre+;Setd2f/f* and control mice to assess the impact of Setd2 loss on the transcriptome of DN3 T cells in an unbiased manner. Gene set enrichment analysis (GSEA) revealed that genes involved in methylated histone binding, G2 DNA damage checkpoint and DNA DSB repair were significantly under-expressed in Setd2 knockout DN3 thymocytes (Fig. 6e). To validate the RNA-seq data, we conducted additional quantitative PCR experiments. Consistent with the above findings, we found that the expression of genes associated with the DNA damage response, including *taok1, dclre1c, mdc1, chek2,* and *prkdc*, was significantly down-regulated in Setd2-deficient thymocytes (Fig. 6f). In addition, immunoblotting analysis showed that activation of the ATM kinase, a major sensor of DSBs, was markedly suppressed in thymocytes upon Setd2 deletion, manifested by the reduced phosphorylation of ATM and its downstream target histone H2AX (Fig. 6g). These data indicate that decreased binding of Rag1 to the TCRβ gene and the impaired DNA damage response after the introduction of Rag1/2-mediated DSBs at RSSs may together lead to the V(D)J gene recombination defect in Setd2 knockout lymphocytes.

**Setd2 is mutated in human primary immunodeficiency.** The defective T and B cell development in Setd2 knockout mouse models is phenotypically reminiscent of that in human primary

immunodeficiency disorder (PID) patients, who suffer from impaired adaptive immunity caused by aberrant function in at least one component of the immune system, such as T and/or B lymphocytes. We then investigated whether genetic alteration of SETD2 was present in PID patients. We analyzed our in-house whole-exon sequencing (WES) data of PID patients who had abnormal function of lymphocytes or immunoglobulin. After eliminating the published polymorphisms in the dbSNP database and the 1000 Genomes human genome polymorphism data set (1000G)[34], we removed variants with a minor allele frequency (MAF) of >0.01 and identified 2 missense SETD2 mutations in PID patents who did not carry previously reported PID-associated genetic alterations[35] (Supplementary Table 6 and Supplementary Fig. 9a). To explore whether these mutations affected SETD2, we constructed expression plasmids with wild-type or mutated SETD2 fused with an HA-tag for reconstitution experiments in a CRISPR/Cas9-mediated Setd2 knockout 293T cell line. Compared to transfection of wild-type SETD2, transfection of the p.K96Q and p.V1070L SETD2 mutants failed to restore the trimethylation of histone H3, suggesting that these mutations impaired the enzymatic activity of SETD2. These data collectively imply an important clinical relevance of SETD2 mutations in the development of PID in humans (Supplementary Fig. 9b, c).

## Discussion

Utilizing multiple genetically engineered mouse models to specifically delete the epigenetic regulator Setd2 in different hematopoietic lineages, we uncovered an essential role for Setd2 in normal lymphopoiesis in the current study. *Mx1-Cre+;Setd2f/f* mice display prominent loss of mature T and B cells and accumulation of CLPs and early T and B cell progenitors. The phenotype of *Mx1-Cre+;Setd2f/f* mice is further substantiated by consistent observations from *Lck-Cre+;Setd2f/f* and *Cd19-Cre+; Setd2f/f* mice in which Setd2 is deleted in the T or B cell lineage, respectively. Further analysis revealed that thymocyte development is blocked at the DN3 stage in Setd2-deficient mice and that Setd2-null B cell differentiation is arrested at the pro-B stage. Interestingly, V(D)J recombination of TCRβ and immunoglobulin, a critical step in lymphopoiesis, occurs at the DN3 or pro-B stage[5,8]. We further showed that trimethylation of H3K36 catalyzed by Setd2 is enriched at the TCRβ and Igh loci, especially in V, D, and J clusters but not in intergenic regions. Setd2 deletion leads to loss of H3K36me3 and severe impairment of V(D)J rearrangement. Therefore, Setd2-mediated histone modification is required for V(D)J recombination and lymphocyte development (Fig. 7).

Two independent groups recently reported that Setd2 is indispensable for the self-renewal of HSCs and differentiation of myeloid as well as erythroid lineages[36,37]: deletion of exon 1 of Setd2 at results in defective hematopoietic-repopulating activity of HSCs and a myelodysplastic syndrome-like phenotype[37]; ablation of exon 6 of Setd2 dramatically decreases the population of HSCs and leads to leukopenia, anemia, erythroid dysplasia, and

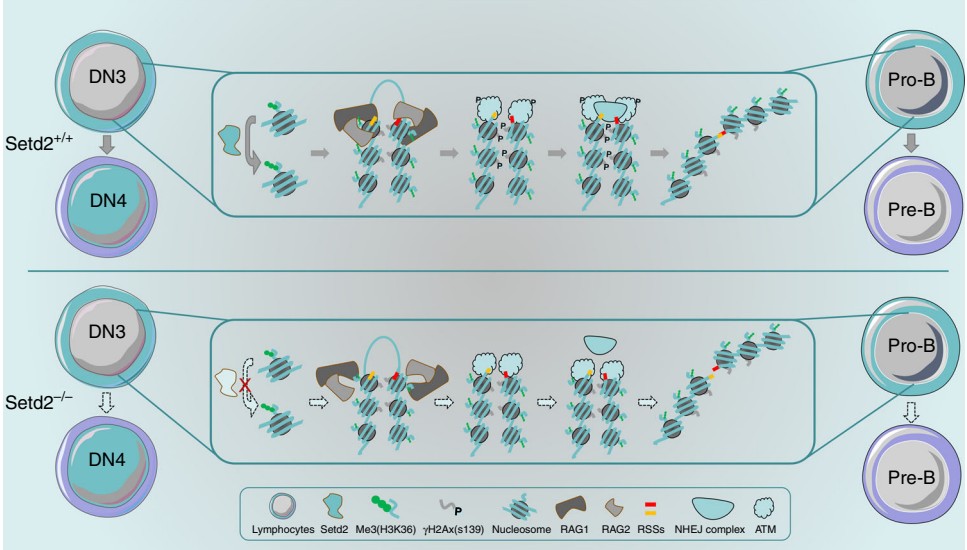

**Fig. 7** Working model for the role of Setd2 in V(D)J rearrangement during early T or B cell development. V(D)J rearrangement is a key event in the generation of diverse antigen receptors and in normal lymphoid development and occurs at the DN3 to DN4 stage in T cells and the pro-B to pre-B stage in B cells. Setd2, a major enzyme that catalyzes the trimethylation of the lysine residue at H3K36, is required for the recruitment of the key recombinase RAG1 to the TCRβ and Igh gene loci and subsequent ATM-mediated DNA damage repair. Setd2 deletion leads to a loss of H3K36me3 and markedly abrogates V(D)J rearrangement of TCRβ and Igh, which causes a severe developmental block at the DN3 and pro-B stages in T and B lymphocytes, respectively

increased thrombopoiesis[36]. In our experiments, we found a similar compromised multilineage reconstitution capacity of Setd2$^{-/-}$ HSCs. We observed a relatively broad expression pattern of Setd2 with preferential expression in hemopoietic stem/progenitor cells and lymphoid cells, including early progenitors of lymphocytes. Therefore, we focused our study of Setd2 on lymphopoiesis. Our analysis of conditional deletion mouse models reveals that leukopenia in Setd2 knockout mice is caused by differentiation arrest at the DN3 or pro-B stage, suggesting an intrinsic requirement for Setd2 in committed lymphopoiesis. Our data, with findings from other groups, demonstrate context-dependent roles of Setd2 in the development of different hematopoietic cell lineages.

An accessible chromatin environment is a prerequisite for V(D)J recombination[16,38,39]. However, the current understanding of epigenetic modifications in the regulation of V(D)J recombination is still very limited. Our results, for the first time, reveal that H3K36me3 is enriched in Vβ, Dβ, and Jβ segments located at or close to the RSSs of individual segments. We further demonstrated that the distribution of H3K36me3 at the TCRβ locus overlaps with the Rag1 binding pattern. Rag1 interacts with H3K36me3, and the occupancy of Rag1 on the TCRβ gene is suppressed after Setd2 loss-mediated H3K36me3 downregulation, suggesting a requirement for H3K36me3 in recruiting Rag1 to the TCRβ locus. A relatively low level of Rag1 enrichment at TCRβ loci was still detected after ablation of Setd2, suggesting that other regulatory factors are involved in the recruitment of Rag1 to RSSs. However, our finding that Setd2 deletion results in severe defects in V(D)J recombination of TCRβ and Igh and, consequently, a marked blockage of lymphocyte development highlights the importance of proper H3K36me3 in V(D)J recombination.

It has been reported previously that H3K4me3 is required for Rag2 recruitment and efficient V(D)J rearrangement[16,17,32]. In this study, we did not detect alterations in Rag2 expression or the H3K4me3 level after Setd2 ablation. H3K4me3 deposition at selected segments of the TCRβ and Igh genes was not affected by Setd2 knockout, suggesting that the H3K36me3 deficiency-induced V(D)J recombination defect did not result from altered H3K4me3. Interestingly, we found that the deposition of

H3K4me3 and H3K36me3 at TCRβ loci showed a distinct pattern. H3K4me3 tended to distribute at J segments, while H3K36me3 enrichment was increased at D and mostly V segments. We also analyzed the correlation between H3K4me3 and H3K36me3 deposition at TCRβ loci using the Pearson correlation coefficient and found a negative correlation ($r = -0.4345$, $p = 0.0011$). We showed that H3K36me3 facilitated the recruitment of Rag1 to RSSs. Taken together, these findings indicate that Rag1-H3K36me3 and Rag2-H3K4me3 may play differential roles at different segments during V(D)J recombination at TCRβ loci.

Setd2 has been demonstrated to be required for DNA DSB repair[21]. We demonstrated in the current study that Setd2 knockout causes impaired activation of ATM, the sensor of DSBs, and phosphorylation of its downstream effector H2AX in thymocytes. This finding is very interesting, considering that Setd2 knockout mice displayed phenotypes reminiscent of those associated with abnormal lymphocyte development in ATM or H2AX mutant animals, including defects in V(D)J rearrangement and aberrant lymphocyte development[40–43]. Therefore, the insufficient DNA damage response in Setd2 knockout DN3 thymocytes could additionally contribute to the V(D)J recombination defect.

We found a greater enrichment of H3K36me3 modifications in V/D/J clusters than in intergenic regions. The precise reason for this phenomenon remains unclear. However, we propose several hypotheses: (1) Previous reports have shown that H3K36me3 deposition occurs preferentially at exons rather than introns and intergenic regions. This preference is evolutionarily conserved[22,39]. (2) H3K36me3 has been reported to be essential for DNA DSB repair[21]. A high level of H3K36me3 at V/D/J clusters will effectively initiate the repair signal for double-strand DNA breaks (DSBs) at RSSs that flank V/D/J segments. (3) H3K36me3 has been shown to be frequently associated with active transcription by participating in pre-mRNA elongation[22]. Successful V(D)J recombination will generate a coding VDJ joint that expresses a functional TCR/BCR, and H3K36me3 may participate in transcriptional elongation[44–46]. These hypotheses await further study.

It is noteworthy that a recent report by Dr. Sandeep S. Dave's group identifies Setd2 as the most frequently mutated gene in

enteropathy-associated T cell lymphoma (EATL), a neoplastic disease derived from γδ T cells. This group found preferential differentiation of the γδ T cell population over the αβ T cell population in Setd2 knockout mice[25], consistent with our observation that αβ T cell differentiation is blocked by Setd2 deletion due to defective TCRβ generation by V(D)J recombination. It is conceivable that the diversity of the TCRγδ is very limited and that V(D)J recombination is not carried out as extensively in γδ T cells as in αβ T cells.

Importantly, our work defines Setd2 as a novel regulator of V(D)J recombination and shows that SETD2 is mutated in human PID patients. The results of the functional assays showed that the SETD2 mutations found in PID patients impair its enzymatic activity to catalyze trimethylation of the lysine residue at H3K36, providing additional support for the clinical relevance of SETD2 mutations, especially those associated with lymphopenia and immunoglobin defects, in PID patients. This study provides new and important insights into the epigenetic regulation of somatic DNA rearrangement in TCRs and immunoglobins, which is critical for the development of the adaptive immune system.

## Methods

**Mice**. The $Setd2^{f/f}$ mouse model was established by gene targeting via homologous recombination in C57BL/6 embryonic stem cells. Loxp sequences were introduced to flank exon 6 and exon 7 of Setd2. $Mx1$-$Cre$, $Cd19$-$Cre$, and $Lck$-$Cre$ mice on a C57BL/6 background were purchased from the Jackson Laboratory. $Setd2^{f/f}$ mice were crossed with specific Cre transgenic mice to achieve lineage-specific gene deletion. All mice were maintained and bred in the pathogen-free facility at Renji Hospital. Mouse experimental protocols were approved by the Renji Hospital Animal Care and Use Committee.

**FACS**. Single-cell suspensions of thymus and spleen were prepared by gently pressing the tissue through a 40 μm BD cell strainer. We harvested bone marrow cells from mouse femurs and tibias and collected peripheral blood from the retroorbital sinus. Red blood cells were lysed using ACK buffer. Nucleated blood cells were then stained for 30 min on ice with the indicated fluorochrome-conjugated antibodies (BD Bioscience and eBiosciences) in PBS containing 0.5% BSA. Flow cytometric sorting or analysis was performed using a FACSAria II (BD Biosciences) or LSR Fortessa (BD Biosciences) platform. FACS data were analyzed with FlowJo X software. The antibodies used in the study and their clone numbers are listed in Supplementary Table 2, the gating strategies are presented in Supplementary Fig. 10.

**Hematoxylin and eosin and IHC staining**. Mouse thymuses were harvested and fixed in 4% PFA for 2 h and were then processed, sectioned at 4 μm and stained with hematoxylin and eosin according to the standard procedures. For bone marrow IHC, mouse femurs were isolated and fixed in 4% PFA for 6 h and decalcified for 24 h (10 g of EDTA-Na$_2$ and 15 mL of 4% formaldehyde in 100 mL, pH 8.0), and were then embedded with paraffin, sectioned at 4 μm. The slides were deparaffinization and processed by heat-induced epitope retrieval in sodium citrate buffer (10 mM sodium citrate, 0.1% Tween 20, pH 6.0), then blocked with 5% BSA-PBST for 2 h and incubate with primary antibodies with 1:200 dilutions for overnight at 4 °C, slides then washed by PBS for 3 times and incubated with secondary antibodies for 2 h at room temperature. The antibodies used are listed in Supplementary Table 2.

**Immunoblotting**. Total BM cells were isolated from the femurs of $Mx1$-$Cre^+$; $Setd2^{f/f}$ mice and control mice at 8 weeks post pIpC treatment. Immunoblotting was conducted using conventional methods described previously[47]. Briefly, the cells were lysed with RIPA Lysis and Extraction Buffer for 1 h at 4 °C, then further lysed by sonication. The cells lysates were mixed with loading buffer for loading in SDS-PAGE gel, and transfer to nitrocellulose filter membrane (NC membrane), the membranes were blocked with 5% BSA-TBST for 1 h at room temperature and incubated with primary antibodies with dilutions for overnight at 4 °C followed by incubating the appropriate secondary antibodies. The antibodies with dilutions used are listed in Supplementary Table 2.

**Bone marrow transplantation assays**. BMNCs were harvested from the femurs and tibias of 8–10-week-old $Mx1$-$Cre^+$;$Setd2^{f/f}$ and control mice. The CD45.1 recipient mice were administered 10 Gy of lethal radiation before bone marrow transplantation. A total of $2 \times 10^6$ donor BMNCs were mixed with BMNCs from CD45.1 mice at a 1:1 ratio and injected into the recipient mice via the tail vein. Four weeks after bone marrow transplantation, the recipients were injected with three doses of pIpC. Beginning 2 weeks after the final injection, peripheral blood was collected monthly from the retroorbital sinus of the recipients for chimerism analysis. Recipient mice were sacrificed at 4 months after pIpC injection for FACS analysis of bone marrow and thymus cells.

**RNA-seq and RT-PCR**. RNA was extracted by a Quick-RNA MicroPrep Kit (ZYMO Research) according to the manufacturer's instructions. RNA-seq was performed by Annoroad Gene Technology Corporation (Beijing) as previously described, and the raw and processed data were deposited in the NCBI Gene Expression Omnibus (GEO) database under accession number GSE116685. First-strand cDNA was synthesized using a PrimeScript 1st strand cDNA Synthesis Kit (TaKaRa Bio). Real-time PCR (RT-PCR) was performed using AceQ Universal SYBR Green qPCR Master Mix (Vazyme Biotech Co., Ltd.). The primers used for RT-PCR are listed in Supplementary Table 3.

**Chromatin immunoprecipitation**. Single thymocyte suspensions were prepared from 8–10-week-old wild-type C57BL/6 mice, followed by antibody staining. CD4$^-$CD8$^-$CD44$^-$CD25$^+$ DN3 thymocytes were sorted by flow cytometry. DN3 thymocytes were crosslinked with 1% formaldehyde for 15 min. ChIP assays were conducted using a SimpleChIP Enzymatic Chromatin IP Kit (Cell Signaling Technology) following the manufacturer's protocol. The ChIP results were quantified by real-time PCR (RT-PCR) using AceQ Universal SYBR Green qPCR Master Mix (Vazyme Biotech). The primers used for the ChIP assay are listed in Supplementary Table 4. For the V, D, and J segments of TCRβ, PCR primers were designed at or close to RSSs. For the intergenic regions, primer sites were selected every 10 kb from TRBV30 to TRBVD1.

**Retroviral transfection**. The detailed procedure for generation of retrovirus-mediated TCR-expressing mice was described in *Nature Protocol*[30]. Briefly, we first constructed a 2 A peptide-linked TCRα/β retroviral plasmid coexpressed GFP. Bone marrow cells were harvested from 5-FU-treated $Lck$-$Cre^+$;$Setd2^{f/f}$ mice, transfected with TCRα/β-expressing retrovirus and transplanted into irradiated recipient mice. Eight weeks after bone marrow transplantation, recipient mice were sacrificed, and the bone marrow and thymus were harvested for genomic DNA purification of GFP$^+$ cells and flow cytometric analysis, respectively.

**Coimmunoprecipitation**. Coimmunoprecipitation of endogenous RAG1 and H3K36me3 from the thymus of 10-week-old mice was performed using CelLytic™ MT Cell Lysis Reagent, the single cell suspensions were lysed for 2 h at 4 °C, then centrifuge the cell lysates and incubate with primary antibodies at 4 °C for 16 h. The dilutions for antibodies: RAG1, 1:30; H3K36me3, 1:100; H3K36me2, 1:100. After then the immunoprecipitated complexes were pulled down with protein A/G beads (Roche) 4 °C for 2 h. The immunoprecipitated complexes were performed by Immunoblotting follow the standard procedure. The antibodies used for immunoprecipitation are listed in Supplementary Table 2.

**PCR assay for V(D)J recombination**. Genomic DNA was extracted from sorted mouse DN3 thymocytes (CD4$^-$CD8$^-$CD44$^-$CD25$^+$), pro-B cells (B220$^+$ CD43$^+$CD19$^+$IgM$^-$), DP thymocytes (CD4$^+$CD8$^+$CD44$^-$CD25$^-$), and primary mouse kidney cells. Genomic DNA was diluted 90, 30, and 10 ng (or 200, 100, and 50 ng for DP thymocytes, 34 cycles or 150, 75, and 30 ng for pro-B cells, 32 cycles) and subjected to 31 cycles of PCR amplification (24 or 25 cycles for CD14). We used upstream primers in V or D segments and reverse primers in J segments for individual PCRs to amplify TCRβ, Igh, or TCRα gene rearrangements. The PCR products were separated by 6–8% denaturing PAGE and visualized by ethidium bromide staining. The primers used to amplify V(D)J joints are shown in Supplementary Table 5.

**Patients**. This study was approved by the ethics committee of the Children's Hospital of Fudan University. Written informed consent was obtained from the parents of all patients. The study was conducted in accordance with the approved guidelines. The patients' clinical features are shown in Supplementary Table 6. Peripheral blood was collected for whole-exon sequencing. These patients were from a nonconsanguineous Chinese family with healthy parents and without a family history of specific diseases. The study is compliant with the "Guidance of the Ministry of Science and Technology (MOST) for the Review and Approval of Human Genetic Resources" in China.

**Whole-exome sequencing and variant analysis**. After the collection of 2 mL of EDTA-anticoagulated blood, DNA was extracted from peripheral blood mononuclear cells, which were isolated using lymphocyte separation medium (Beyotime Biotechnology, Shanghai, China). Fragments of patients' genomic DNA were enriched for panel sequencing using an Agilent ClearSeq Inherited Disease panel kit. Enriched DNA samples were indexed and sequenced on a HiSeq 2000 sequencer (Illumina, San Diego, CA) using standard protocols in a Clinical Laboratory Improvement Amendments (CLIA)-compliant sequencing laboratory at WuXi NextCODE (CLIA ID 99D2064856). Sequencing provided at least 50 million 125-bp paired-end reads for each sample.

Reads were mapped to the human reference genome hg19 using Burrows–Wheeler Aligner (BWA). The average coverage sequencing depth for official targets was at least 180× and was higher than 20× for >99% of the targeted region. Variants were called in accordance with the gold standard of the GATK Best Practices. Deleterious mutations and novel variants detected using WES were confirmed via Sanger sequencing. A coverage-based algorithm, CANOES, was used to detect large exonic deletions and duplications. Based on the validation results of the array comparative genomic hybridization (CGH) experiments, ratios of less than 0.65 and greater than 1.35 were scored as deletions and duplications, respectively. All positive calls were further investigated and confirmed via Sanger sequencing.

**Plasmids**. The SETD2-HA plasmid was a gift from Professor Min Wu at Wuhan University. Mutated plasmids were constructed by high-fidelity PCR-based specific mutation using SETD2-HA as a template, and mutations were confirmed via Sanger sequencing.

**Statistical analysis**. All statistical analysis for flow cytometric data was performed with GraphPad 7.0 software using Student's $t$ test assuming equal variance and considering $p < 0.05$ significant. The Pearson correlation coefficient (Pearson $r +1$ to $-1$) was used to measure the strength of the correlation between conditions for the ChIP assay results[31].

**Reporting summary**. Further information on research design is available in the Nature Research Reporting Summary linked to this article.

## Data availability
The authors declare that the data supporting the findings of this study are available within the Article, Supplementary Information files, and Source Data, or are available upon reasonable requests to the authors. RNA-seq data are deposited in NCBI Gene Expression Omnibus (GEO) database under accession number GSE116685.

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

## Acknowledgements

The study was supported by funds to W. Gao from the National Key R&D Program of China (2017YFA0102900), the National Natural Science Foundation of China (NSFC, 81630073 and 81872406), the Science and Technology Commission of Shanghai Municipality (16JC1405700), the Education Commission of Shanghai Municipality (for the High Peak IV Subject on Stem Cells and Translational Medicine Research) and the KC Wong foundation; and by funds to H.H. Zhu from the NSFC (81772743), Shanghai Municipal Education Commission–Gaofeng Clinical Medicine Grant Support (20181706), Shanghai Rising-Star Program (17QA1402100), the Shanghai Youth Talent Support Program, School of Medicine, Shanghai Jiao Tong University (Excellent Youth Scholar Initiation Grant 16XJ11003), and the Innovative Research Team of High-level Local Universities in Shanghai; and by funds to L. Li from the NSFC (81772938), the Science and Technology Commission of Shanghai Municipality (18140902700, 19140905500), State Key Laboratory of Oncogenes and Related Genes (KF01801) and the Innovation Research Plan supported by the Shanghai Municipal Education Commission (ZXGF082101). The authors thank professor Fubin Li from Shanghai Institute of Immunology for sharing his expertise, and the anonymous contributor from the https://smart.servier.com/ for sharing the partial materials for making schematic diagram of this paper.

## Author contributions

Z.J., H.H.Z., and W.G. designed the experiments, analyzed the data and wrote the manuscript. Z.J. performed the mouse studies, immunoblotting, RNA-seq analysis, and FACS. Y.S., J.W., and H.Z. contributed to the bone marrow transplantation and generated the SETD2 KO cell line. C.C., X.W., and K.L. contributed to plasmid construction. X.L., L.X., and J.Y. provided support for the animal experiments. K.Z., J.M., C.C., J.H., and L.S. contributed to the data analysis. L.L. provided the mouse model and contributed to data analysis. W.Z. and J.S. collected the patient samples, performed WES, and analyzed the data.

## Additional information

**Competing interests:** The authors declare no competing interests.

