## [Peer Review File · Nature Communications]

Reviewers' comments:

Reviewer #1 (B/T development, epigenetic regulation)(Remarks to the Author):

The authors identify and characterize a function for Setd2, a histone methyltransferase that mediates the trimethylation of lysine 36 on histone 3 (H3K36me3), in the maintenance of HSC, abundance of CLP, and the development of T and B cell lineages. Using conditional deletion of Setd2 in hematopoietic stem cells and T-cell lineages, they show that T- and B-cell development is blocked at the DN3 and the pro-B cell stages respectively. The authors propose that these developmental defects indicate a role for Setd2 in V(D)J recombination involving the TCR β locus and the Igh locus. A potential mechanism is proposed in which H3K36me3 mediates the recruitment of Rag1. Other alterations of Setd2 deletion in mice are suggested, impaired DNA double-strand breaks repair system, that could play a role in this phenotype. They finally identified potential consequences of Setd2 polymorphisms in the development of human primary immunodeficiency.

This is an interesting paper and of significant interest. A number of issues need to be addressed to further improve the manuscript.

Major concerns.

1. The use of a B-cell specific promoter (i.e. CD19-cre or Mb1-cre) driving CRE expression crossed with the Setd2 fl/fl mice is needed to specifically address the role of Setd2 in B-cell development.
2. The data supporting the specificity for effects of V(D)J recombination in B cells is incomplete (H3K36me3 and RAG1 ChIP in the Igh loci).
3. Fig 5. ChIP on H3K36me3 needs to be done for Setd2^{-/-} cells to compare the background levels of this modification.
4. Is the recruitment of Rag1 by H3K36me3 specific? From the data in fig 6C, it seems that Rag1 recruitment is affected to various degrees or not at all.
5. Statistical analysis for correlation between H3K36me3 and Rag1 deposition needs to be done.
6. Recruitment of Rag1 by H3K4me3 is well established. Control ChIP experiments are required to address if Rag1 binding changes in Setd2^{-/-} mice are caused by a reduction in H3K4me3 levels.
7. Figure legends need better description instead of stating the results.

Minor concerns.

1. The text needs work. There are several typos, multiple grammatical, and some sentences are confusing. The introduction has some references that are not up to date.
2. It is well established that Igh locus rearrange at the pro-B cell stage. The authors repeatedly mention the Igh rearrangement is impaired at the pre-B cell stage.
3. Figures need more clarity and better labeling.
4. Display of the ChIP results in Fig5 and 6 is not ideal in order to see the changes in enrichment between conditions.
5. Legend for Figure 4d is missing.
6. Legend for Figure S3d is missing.

Reviewer #2 (Thymocyte development, VDJ recombination)(Remarks to the Author):

SetD2 is the only known histone H3K36me3 methyltransferase. In this manuscript, Ji et al found that Setd2 deletion leads to a loss of H3K36me3 and markedly suppresses V(D)J rearrangement presumably due to the impaired binding of RAG1 to the TCR β gene and DNA DSB repair. As a result, the intrathymic T cell development is blocked at DN3 stage. The observation of this manuscript is interesting and the experiments are clearly described in general. However, many questions remain to be addressed:

Major revision:

1. It had been reported that trimethylated histone H3K4 is required for the binding of RAG2 and the high efficiency of gene rearrangement, the roles of the trimethylation of H3K4 and H3K36 in V(D)J recombination (rather than binds to Rag2 and Rag1, respectively) should be compared and discussed.
2. Why SetD2 selectively trimethylates the V/D/J clusters but not the intergenic regions?
3. Clearly there is a block at the transition from DN3 to DN4 in the absence of Setd2, but not obvious from DP to SP transition; why SetD2 deficiency has no impact on the TCR α rearrangement?
4. SETD2 is responsible for the trimethylation of lysine 36 of Histone H3 in general. To verify whether SetD2 indeed mainly functions on V(D)J recombination, a TCRTg to rescue the development block is required.

Minor revision:

1. H3K36Me2 should be used as control rather than IgG in Figure 6D.
2. Please label the ratio (percentage) within each gate.

Reviewers' comments:

Reviewer #1 (B/T development, epigenetic regulation)(Remarks to the Author):

The authors identify and characterize a function for Setd2, a histone methyltransferase that mediates the trimethylation of lysine 36 on histone 3 (H3K36me3), in the maintenance of HSC, abundance of CLP, and the development of T and B cell lineages. Using conditionally deletion of Setd2 in hematopoietic stem cells and T-cell lineages, they show that T- and B-cell development is blocked at the DN3 3 and the pro-B cell stages respectively. The authors propose that these developmental defects indicate a role for Setd2 in V(D)J recombination involving the TCR β locus and the Igh locus. A potential mechanism is proposed in which H3K36me3 mediates the recruitment of Rag1. Other alterations of Setd2 deletion in mice are suggested, impaired DNA double-strand breaks repair system, that could play a role in this phenotype. They finally identified potential consequences of Setd2 polymorphisms in the development of human primary immunodeficiency.

This is an interesting paper and of significant interest. A number of issues need to be addressed to further improve the manuscript.

Major concerns.

1. The use of a B-cell specific promoter (i.e.CD19-cre or Mb1-cre) driving CRE expression crossed with the Setd2 fl/fl mice is needed to specifically address the role of Setd2 in B-cell development.

Reply: We thank the reviewer very much for the precise summary of our manuscript and positive comments of our work. Following the reviewer's advice, we crossed the *Cd19-Cre* mice with the *Setd2^{fl/fl}* mice to specifically knockout Setd2 in B cell. As shown in **supplementary Fig.6** of the revised manuscript, we found that specific ablation of Setd2 in the B cell lineage resulted in a developmental block at the pro-B stage, phenotypically consistent with the B cell defects in the *Mx1-Cre⁺;Setd2^{fl/fl}* mouse model as shown by us previously. We also performed semi-quantitative PCR assays to examine the V(D)J rearrangement from sorted pro-B cells of *Cd19-Cre⁺;Setd2^{fl/fl}* and control mice. As shown in **supplementary Fig.6c**, the V(D)J recombination was impaired in Setd2 null B cells.

2. The data supporting the specificity for effects of V(D)J recombination in B cells is incomplete (H3K36me3 and RAG1 ChIP in the Igh loci).

Reply: To address this concern, we have performed new ChIP assays of H3K36me3 and Rag1 at Igh loci in sorted B cells from control and *Cd19-Cre⁺;Setd2^{fl/fl}* mice. We found that the deposition of H3K36me3 and Rag1 showed an overlapping pattern, similar to what we observed at TCR β loci. ChIP experiments results have been incorporated to the revised manuscript as **supplementary Fig. 7**.

3. Fig 5. ChIP on H3K36me3 needs to be done for Setd2^{-/-} cells to compare the background levels of this modification.

Reply: As suggested, we performed additional ChIP experiments on sorted Setd2 null B and T

cells to determine the background levels of H3K36me3. We can barely detect enrichment of H3K36me3 either at IgH or TCR β loci. The new data has been included to **supplementary Fig.2a** of the revise manuscript.

4. Is the recruitment of Rag1 by H3K36me3 specific? From the data in fig 6C, it seems that Rag1 recruitment is affected to various degrees or not at all.

Reply: We greatly appreciated the reviewer's insightful comment. The V(D)J rearrangement is accurately regulated by multiple factors. In our manuscript, we found that Setd2 ablation results in removal of the H3K36me3 modification and decreased (but not complete ablation of) Rag1 recruitment to RSSs during V(D)J rearrangement. Therefore, we do not exclude other factors besides H3K36me3 in contributing to the recruitment of Rag1 to the RSSs. We have added a short discussion on this point in line 382 to 390 of the revised manuscript text.

5. Statistical analysis for correlation between H3K36me3 and Rag1 deposition needs to be done.

Reply: Following the reviewer's advice, we have used the Pearson's Correlation Coefficient method to measure the correlation between H3K36me3 and Rag1 deposition at RSSs. We found that the correlation r value is 0.7898, and the p value is 0.010 between H3K36me3 and Rag1 deposition on TCR β locus, which indicates a strong positive correlation.

6. Recruitment of Rag2 by H3K4me3 is well established. Control ChIP experiments are required to address if Rag1 binding changes in Setd2 $^{-/-}$ mice are caused by a reduction in H3K4me3 levels.

Reply: We thank the reviewer's expert view. According to the reviewer's advice, we performed new immunoblotting experiments to examine the H3K4me3 level in Setd2 $^{+/+}$ and Setd2 $^{-/-}$ lymphocytes. As shown in **supplementary Figure. 8b, c** of the revised manuscript, Setd2 deletion did not cause alterations in H3K4me3 and Rag2 (**Fig5a,b** and **supplementary Fig. 8a**). In addition, we performed ChIP assays of H3K4me3 at TCR β and Igh loci, and did not detect significant reduction of H3K4me3 depositions at TCR β and Igh genes by Setd2 knockout. This data is incorporated to the revised manuscript as **supplementary Figure. 8**. Collectively, these data suggest that the aberrant V(D)J rearrangement in Setd2 deficient lymphocytes is caused by ablation of H3K36me3 without affecting H3K4me3.

7. Figure legends need better description instead of stating the results.

Reply: We thank the reviewer for the careful reading of our manuscript, we have expanded the figure legends and included more detailed description in the revised manuscript.

Minor concerns.

1. The text needs work. There are several typos, multiple grammatical, and some sentences are confusing. The introduction has some references that are not up to date.

Reply: We have made amendments according the reviewer's suggestion. The manuscript text has also been edited by American Journal Experts company. We have updated the references in the

revised manuscript.

2. It is well established that Igh locus rearrange at the pro-B cell stage. The authors repeatedly mention the Igh rearrangement is impaired at the pre-B cell stage.

Reply: We greatly appreciated the reviewer's expert comment. When we looked again at our data, the most prominent blockage of B cell development in the Setd2 knockout mice was found at the pro-B stage. This is consistent with what the reviewer pointed out that Igh rearrange occurs at the pro-B cell stage. We have modified manuscript accordingly.

3. Figures need more clarity and better labeling.

Reply: Probably due to the fact that we combined the manuscript and the figures in one PDF file for the first submission, the file converting process decreased the image resolution. Following the advice, we have replaced with high resolution figures for this submission. We have also made amendments to the figure labeling to make it clearer.

4. Display of the ChIP results in Fig5 and 6 is not ideal in order to see the changes in enrichment between conditions.

Reply: As suggested, we modified the way we present the ChIP results by comparing between conditions at the individual segments for better illustration in revised **Figure 5 and 6**.

5. Legend for Figure 4d is missing.

Reply: We apologized for the carelessness. We have added the figure legend to figure 4d. Thanks for the reviewer's careful reading our manuscript.

6. Legend for Figure S3d is missing.

Reply: We have added the figure legend to figure S3d. Thanks again for the reviewer's careful reading our manuscript.

Reviewer #2 (Thymocyte development, VDJ recombination)(Remarks to the Author):

SetD2 is the only known histone H3K36me3 methyltransferase. In this manuscript, Ji et al found that Setd2 deletion leads to a loss of H3K36me3 and markedly suppresses V(D)J rearrangement presumably due to the impaired binding of RAG1 to the TCR β gene and DNA DSB repair. As a result, the intrathymic T cell development is blocked at DN3 stage. The observation of this manuscript is interesting and the experiments are clearly described in general. However, many questions remain to be addressed:

Reply: We thank the reviewer very much for the summary and inspiring comments.

Major revision:

1. It had been reported that trimethylated histone H3K4 is required for the binding of RAG2 and the high efficiency of gene rearrangement, the roles of the trimethylation of H3K4 and H3K36 in V(D)J recombination (rather than binds to Rag2 and Rag1, respectively) should be compared and discussed.

Reply: We are grateful for the expert comments and suggestions for the reviewer. Accordingly, we have performed ChIP assays on H3K4me3 at TCR β gene, we found that the deposition of H3K4me3 and H3K36me3 at TCR β loci showed a distinct pattern (**Fig 5a and supplementary Fig. 8b**). H3K4me3 tended to distribute at J segments, while H3K36me3 was more enriched at D and mostly V segments. We also analyzed the correlation between H3K4me3 and H3K36me3 deposition using Pearson's Correlation Coefficient method at TCR β loci and found a negative correlation ($r = -0.4345$, $p = 0.0011$). Previous reports demonstrated the interaction between Rag2 and H3K4me3 is required for the V(D)J recombination¹⁻⁴. In our manuscript, we showed that the H3K36me3 facilitated recruitment of Rag1 to RSSs. Taken together, those findings indicate that the Rag1-H3k36me3 and Rag2-H3K4me3 might take differential roles at different segments during V-D-J recombination at TCR β loci.

In addition, we performed new immunoblotting experiments to examine the H3K4me3 level in *Setd2*^{+/+} and *Setd2*^{-/-} lymphocytes. As shown in **supplementary Figure 8a** of the revised manuscript, *Setd2* deletion did not cause alterations in H3K4me3. Moreover, we performed ChIP assays of H3K4me3 at *Igh* loci, and did not detect significant reduction of H3K4me3 depositions at TCR β and *Igh* genes by *Setd2* knockout (**supplementary Figure 8b,c** of the revised manuscript), suggesting that the H3K36me3- deficiency-caused VDJ recombination defect was not resulted from altered H3K4me3.

2. Why *SetD2* selectively trimethylates the V/D/J clusters but not the intergenic regions?

Reply: We thank the reviewer's insightful question. As correctly pointed out by the reviewer, we did find a more enrichment of H3K36me3 modifications in V/D/J clusters than in intergenic regions. The precise reason beneath this phenomenon remains unclear. However, there are several hypotheses from us: 1) Previous reports have shown that the H3K36me3 preferentially deposits at exons relative to the introns and intergenic regions. This preference is evolutionarily conserved^{5,6}. 2) H3K36me3 has been reported to be essential for the DNA DSB repair⁷. High level of H3K36me3 at V/D/J clusters will effectively initiate the repair signal of double strand DNA breaks (DSB) at RSSs that flank V/D/J segments. 3) H3K36me3 has been shown to be frequently associated with active transcription by participating in pre-mRNA elongation⁵. Successful V(D)J recombination will generate a coding V-D-J joint that express a functional TCR/BCR, the H3K36me3 may participate in the transcriptional elongation⁸⁻¹⁰. We have added a paragraph of discussion on these points in the revised manuscript (line 415-425).

3. Clearly there is a block at the transition from DN3 to DN4 in the absence of *Setd2*, but not obvious from DP to SP transition; why *SetD2* deficiency has no impact on the TCR α

rearrangement?

Reply: We greatly appreciate the expert question. As we shown in **figure3 d, e**, the relative ratio of SP versus DP was 0.35 in *Setd2^{fl/fl}* while the ratio was 0.22 in *Setd2* deficiency mice, indicating that the ablation of *Setd2* also caused a moderate blockage of the transition from DP T cells to SP T cells. To assess whether TCR α recombination is also affected by the loss of *Setd2*, we sorted the DP thymocytes for V(D)J rearrangement assays at TCR α loci. We found that *Setd2* deletion also led to an impairment of V(D)J recombination at TCR α , although the effect on TCR α recombination was not as profound as on the TCR β gene recombination. These data are now provided as **supplementary Figure 3a** in the revised manuscript.

4. SETD2 is responsible for the trimethylation of lysine 36 of Histone H3 in general. To verify whether *Setd2* indeed mainly functions on V(D)J recombination, a TCRTg to rescue the development block is required.

Reply: Following the reviewer's advice, we obtained OTII TCR transgenic mice that insert the arranged TCR in Y chromosome, and crossed them with the *Lck-Cre⁺;Setd2^{fl/fl}* line for a rescue experiment. Due to the extremely long time to breed and obtain the TCRtg⁺;Cre⁺;Setd2^{fl/fl} mouse line, We took an alternative approach to generated a retrogenic TCR mouse model by introducing TCR to BM cells via retroviral transfection¹¹. This method was published in *Nature Protocol* and widely used by different groups¹²⁻¹⁴. As shown in **supplementary Figure 4** and described in line 234- 245 of the revised manuscript, we found that expression of TCR in *Setd2^{-/-}* T cell significantly abrogated the developmental blockage of DN3 to DN4 caused by *Setd2* deletion. These data provide further support to the conclusion that *Setd2* is required for V(D)J recombination in normal lymphocyte development.

Minor revision:

1. H3K36Me2 should be used as control rather than IgG in Figure 6D.

Reply: As suggested, we have re-run the Co-IP experiments using the H3K36me2 as control, and updated Fig.6d of the revised manuscript.

2. Please label the ratio (percentage) within each gate.

Reply: As suggested, we have added the ratio for each gate of flow cytometry plots in the revised manuscript.

1. Matthews, A. G. W. *et al.* RAG2 PHD finger couples histone H3 lysine 4 trimethylation with V(D)J recombination. *Nature* **450**, 1106–1110 (2007).
2. Bettridge, J., Na, C. H., Pandey, A. & Desiderio, S. H3K4me3 induces allosteric conformational changes in the DNA-binding and catalytic regions of the V(D)J recombinase. *Proc. Natl. Acad. Sci.* **114**, 1904–1909 (2017).
3. Liu, Y., Subrahmanyam, R., Chakraborty, T., Sen, R. & Desiderio, S. A Plant Homeodomain in Rag-2 that Binds Hypermethylated Lysine 4 of Histone H3 Is Necessary for Efficient Antigen-Receptor-Gene Rearrangement. *Immunity* **27**, 561–571 (2007).
4. Shimazaki, N. & Lieber, M. R. Histone methylation and V(D)J recombination. *Int. J. Hematol.* **100**, 230–237 (2014).
5. Guo, R. *et al.* BS69/ZMYND11 Reads and Connects Histone H3.3 Lysine 36 Trimethylation-Decorated Chromatin to Regulated Pre-mRNA Processing. *Mol. Cell* **56**, 298–310 (2014).
6. Kolasinska-Zwierz, P. *et al.* Differential chromatin marking of introns and expressed exons by H3K36me3. *Nat. Genet.* **41**, 376–381 (2009).
7. Carvalho, S. *et al.* SETD2 is required for DNA double-strand break repair and activation of the p53-mediated checkpoint. *Elife* **2014**, (2014).
8. Luco, R. F. *et al.* Regulation of alternative splicing by histone modifications. *Science* (80-.). (2010). doi:10.1126/science.1184208
9. Carvalho, S. *et al.* Histone methyltransferase SETD2 coordinates FACT recruitment with nucleosome dynamics during transcription. *Nucleic Acids Res.* **41**, 2881–2893 (2013).
10. Venkatesh, S. & Workman, J. L. Set2 mediated H3 lysine 36 methylation: regulation of transcription elongation and implications in organismal development. *Wiley Interdiscip. Rev. Dev. Biol.* **2**, 685–700 (2013).
11. Holst, J. *et al.* Generation of T-cell receptor retrogenic mice. *Nat. Protoc.* **1**, 406–417 (2006).
12. Collison, L. W. *et al.* The inhibitory cytokine IL-35 contributes to regulatory T-cell function. *Nature* (2007). doi:10.1038/nature06306
13. Yang, S. *et al.* Development of optimal bicistronic lentiviral vectors facilitates high-level TCR gene expression and robust tumor cell recognition. *Gene Ther.* (2008). doi:10.1038/gt.2008.90
14. Cipolletta, D. *et al.* PPAR- γ is a major driver of the accumulation and phenotype of adipose tissue T reg cells. *Nature* (2012). doi:10.1038/nature11132

REVIEWERS' COMMENTS:

Reviewer #1 (Remarks to the Author):

the authors have addressed my comments. The manuscript has much improved.

Reviewer #2 (Remarks to the Author):

The authors have addressed all my concerns and made an impressive effort revising the manuscript. I think the work is ready for publication.